# OpenReview forum: "LFQ: Logit-aware Final-block Quantization for Boosting the Generation Quality of Low-Bit Quantized LLMs"
_ICLR.cc/2026/Conference — Submitted to ICLR 2026_

### Official Review · Reviewer_p8dA · 2025-10-31

**Soundness:** 3
**Presentation:** 3
**Contribution:** 3
**Rating:** 4
**Confidence:** 3

**Summary:**

This paper proposes LFQ that solves two critical problems previous papers hasn't noted: previous methods don't optimize the LM-head and over-reliance on MSE loss. The author notes we should optmize the LM-head with the objective of cross-entropy loss. Author proposed a new optmization objective and adapt it different PTQ algorithms. Many empirical experiments show that this method can improve the performance on different benchmarks across different models.

**Strengths:**

- The paper tackles the important problem of quantizaton and proposes a solution that is adaptable to different algorithm
- This paper conducts empirical experiments across many different models and benchmarks.
- The motivation of using cross-entropy instead of MSE is well-established.

**Weaknesses:**

- As discussed in the sufficiency of quantizing solely the final block section, I wonder why applying not applying this to more layers would yield better performance? Could you try more layers(say last 10?) and provide more analysis?
- LM head optimization produces minimal Gain? As the paper noted, many methods neglect the LM head, which takes relative small amount of memory but could critically influence the performance. Though this paper proposes a better method to optimize the LM head quantization, does it worth it to quantize it instead of leaving it in bf16 for better performance if we can better optimize [XWq-XW_fp|_f? I see the paper discuss it in section 2 and Table3, but I don't fully understand the importance of quantizing the LM head as well.

**Questions:**

- Although this paper includes many different ways of quantization, the experiments with AWQ, which is the mainstream quantization method, is not included. I wonder whether authors can provide more experiments with AWQ to show the performance of this methods.
- My main concern is adaption to AWQ and why do we need quantized LM-head(I agree if we have to quantize LM-head, cross-entropy would be a better option)

---

> ### Author Response · Authors · 2025-11-21
> **Dear Reviewer p8dA, [1]**
>
> Dear Reviewer p8dA,
>
> We truly appreciate your insightful and valuable feedback.
>
> --------
>
> $\textbf{[Weakness 1. How about applying LFQ to more Transformer blocks (e.g., last 10 blocks)?]}$
>
> Thank you for the insightful comment. Since our experiments are conducted on a single GPU, as is common in prior PTQ studies, applying LFQ to more than three blocks leads to GPU out-of-memory issues. For this reason, we vary *”the number of topmost Transformer blocks optimized with LFQ (denoted by $k$)”* only up to three  in Figure 2. As shown in Figure 2, the average scores on IFEval and GSM8K remain nearly constant as $k$ increases, and we therefore conclude that applying LFQ to only the final block would be sufficient.
>
> To emphasize that **to which** block LFQ is applied is far more important than **how many** blocks are optimized with LFQ, we conduct the following experiment for Llama 3.1 8B Instruct where we explicitly exclude the final block from LFQ optimization.
> When k=2, instead of applying LFQ to the last two blocks, we apply LFQ only to the second-to-last block. For the last block, we only minimize $\|\|\mathbf{X}\mathbf{W}\_{\text{FP}} - \mathbf{X}\mathbf{W}\_{q}\|\|^2_F$ without using either the LM head or cross-entropy (denoted as “$k=2$ except the last block”).
> Similarly, when $k=3$, we apply LFQ sequentially to the third- and second-to-last blocks, and again, for the last block only, we minimize the loss without employing the LM head or cross-entropy. We then compare these settings with the original $k=2$ and $k=3$ configurations in Figure 2.
>
>
> \<Table A. Comparison of $k=2$ and $k=3$ except the last block with the original $k=2$ and $k=3$ configurations in Figure 2. “LFQ@Last” indicates whether LFQ is applied to the last block. Similarly, “LFQ@Last-1” and “LFQ@Last-2” indicate whether LFQ is applied to the second-to-last and third-to-last blocks, respectively.\>
>
> | Llama 3.1 8B Instruct | LFQ@Last | LFQ@Last-1 | LFQ@Last-2 | MMLU | (IFEval+GSM8K)/2 |
> | :--- | :---: | :---: | :---: | :---: | :---: |
> | FlexRound | X | X | X | $66.19$ | $75.80$ |
> | + $k=2$ except last block | X | O | X | $66.97$ | $76.17$ |
> | + $k=2$ (Fig. 2) | **O** | O | X | $\mathbf{66.99}$ | $\mathbf{77.13}$ |
> | + $k=3$ except last block | X | O | O | $66.98$ | $76.15$ |
> | + $k=3$ (Fig. 2) | **O** | O | O | $\mathbf{67.03}$ | $\mathbf{76.78}$ |
> | | | |
> | OmniQuant | X | X | X | $64.87$ | $74.39$ |
> | + $k=2$ except last block | X | O | X | $65.29$ | $74.47$ |
> | + $k=2$ (Fig. 2) | **O** | O | X | $\mathbf{65.40}$ | $\mathbf{75.32}$ |
> | + $k=3$ except last block | X | O | O | $\mathbf{65.29}$ | $74.65$ |
> | + $k=3$ (Fig. 2) | **O** | O | O | $65.25$ | $\mathbf{75.30}$ |
> ||||
> | Block-AP | X | X | X | $63.24$ | $71.21$ |
> | + $k=2$ except last block | X | O | X | $\mathbf{63.78}$ | $71.30$ |
> | + $k=2$ (Fig. 2) | **O** | O | X | $63.57$ | $\mathbf{71.76}$ |
> | + $k=3$ except last block | X | O | O | $\mathbf{63.81}$ | $71.24$ |
> | + $k=3$ (Fig. 2) | **O** | O | O | $63.64$ | $\mathbf{71.52}$ |
> ||||
>
> As shown in Table A, the MMLU score remains nearly unchanged regardless of whether LFQ is applied to the final block. In contrast, when LFQ is not applied to the final block, the average of IFEval and GSM8K (i.e., “(IFEval+GSM8K)/2”) consistently drops, approaching the performance level of each underlying PTQ technique. These results indicate that for improving the generation quality of low-bit quantized LLMs, it is far more critical to apply LFQ to the final block than to simply increase the number of blocks optimized with LFQ.
>
> We really appreciate your invaluable comment and have revised our manuscript to include this point and the above experimental results in Appendix D.
>
> ---------
>
> $\textbf{[Weakness 2. Importance of quantizing the LM head.]}$
>
> Thank you for raising this concern. We would like to kindly remind the reviewer that we do not quantize the LM head and leave it in BF16, following mainstream quantization works [1, 2, 3]. The key focus of LFQ is on quantizing the main body of the LLM in a LM head-aware fashion, not on how to quantize the LM head itself. Thus, although we incorporate the LM head into the LFQ optimization objective as in Eq. (2), i.e., $\mathcal{L}\_{\text{CE}} (\mathbf{X} \mathbf{W}\_{\text{FP}} \mathbf{W}\_{\text{Head}}, \mathbf{X} \mathbf{W}\_{q} \mathbf{W}\_{\text{Head})}$), we minimize Eq. (2) only with respect to $\mathbf{W}\_{q}$. This means that the LM head $\mathbf{W}\_{\text{Head}}$ is kept frozen. In other words, the LM head is included in the objective to provide an enhanced optimization signal when quantizing the final block. Thus, it is treated as a constant matrix and is not optimized in our method, LFQ.
>
> [1] GPTQ: Accurate Post-Training Quantization for Generative Pre-trained Transformers, ICLR 2023.
>
> [2] Optimize Weight Rounding via Signed Gradient Descent for the Quantization of LLMs, EMNLP 2024 Findings.
>
> [3] ParetoQ: Improving Scaling Laws in Extremely Low-bit LLM Quantization, NeurIPS 2025.
>
> -------

---

> > ### Author Response · Authors · 2025-11-21
> > **Dear Reviewer p8dA, [2]**
> >
> > $\textbf{[Question 1. How about experiments with AWQ?]}$
> >
> > Thank you for the helpful comment. As the reviewer mentioned, AWQ is one of the mainstream quantization techniques. However, as explained in Section 1, layer-wise PTQ techniques such as AWQ generally underperform block-wise PTQ due to lack of backpropagation during optimization, as demonstrated by several existing block-wise PTQ studies [1,2,3]. For this reason, we mainly focus on block-wise PTQ in this paper. For reference, we compare the experimental results of AWQ with those of block-wise PTQ methods in Appendix F.
> >
> > Moreover, since layer-wise PTQ techniques including AWQ do not rely on back-propagation, it is challenging to apply LFQ to AWQ: specifically, it is non-trivial to minimize $\mathcal{L}\_{\text{CE}} (\mathbf{X} \mathbf{W}\_{\text{FP}} \mathbf{W}\_{\text{Head}}, \mathbf{X} \mathbf{W}\_{q} \mathbf{W}\_{\text{Head})}$ with respect to $\mathbf{W}\_{q}$ without gradient updates. Extending LFQ to layer-wise PTQ methods is therefore an interesting direction for future work.
> >
> >
> > [1] OmniQuant: Omnidirectionally Calibrated Quantization for Large Language Models, ICLR 2024.
> >
> > [2] Optimize Weight Rounding via Signed Gradient Descent for the Quantization of LLMs, EMNLP 2024 Findings.
> >
> > [3] LRQ: Optimizing Post-Training Quantization for Large Language Models by Learning Low-Rank Weight-Scaling Matrices, NAACL 2025.
> >
> > ---------
> >
> > Once again, we sincerely appreciate your time and efforts in reviewing our paper. If you have any remaining issues or concerns, please do not hesitate to bring them to our attention.

---

> > > ### Author Response · Authors · 2025-11-27
> > >
> > > Dear Reviewer p8dA,
> > >
> > > We hope this message finds you well.
> > >
> > > We would like to sincerely thank you once again for the time and effort you have devoted to reviewing our paper. With about one week remaining before the deadline for reviewer–author discussions, we kindly wish to draw your attention to our response to your thoughtful comments. If you have any further questions or concerns, please do not hesitate to let us know.
> > >
> > > Best regards,
> > >
> > > Authors

---

> ### Author Response · Authors · 2025-12-02
>
> -------------------------
>
> $\textbf{[Additional experiments on MoE architectures]}$
>
> We conducted additional LFQ experiments on **Qwen3-30B-A3B-Instruct-2507**, one of the most powerful open-source MoE models, to highlight that LFQ is effective for both MoE models and traditional dense models.
>
> \<Table C. Performance of Qwen3-30B-A3B-Instruct-2507 with LFQ under block-wise PTQ (FlexRound). Within the block-wise PTQ setting, the best accuracy is shown in $\textbf{bold}$. “W4g128” denotes 4-bit group-wise quantization with a group size of 128. For Pass@1 evaluation on AIME$^{\prime}$25, we use a temperature of 0.6 and a top-p of 0.95, and sample 16 responses per question with a maximum generation length of 32768 tokens.\>
>
> | Method | # Bits | WikiText2 | AIME$^{\prime}$25 (Pass@1) |
> |:----------|:------:|:-----------:|:-----------:|
> | Qwen3-30B-A3B-Instruct-2507 | BF16 | $7.00$ | $62.50$ |
> | GPTQ | W4g128 | $7.32$ | $53.75$ |
> | Block-wise PTQ | W4g128 | $7.34$ | $55.21$ |
> | Block-wise PTQ + LFQ (Ours) | W4g128 | $\mathbf{7.29}$ | $\mathbf{59.38}$ |
> ||||
>
> The table above demonstrates that LFQ increases the Pass@1 score on AIME$^{\prime}$25 by more than 4 percentage points, thereby substantially reducing the performance gap between the quantized MoE model and its FP counterpart. In addition, we include GPTQ as a baseline and observe that block-wise PTQ typically outperforms layer-wise PTQ approaches such as GPTQ.
>
> -------------------------

---

### Official Review · Reviewer_nk9F · 2025-11-01

**Soundness:** 2
**Presentation:** 2
**Contribution:** 2
**Rating:** 4
**Confidence:** 4

**Summary:**

The paper proposes Logit-aware Final-block Quantization (LFQ) to enhance the performance of quantized LLMs. Unlike conventional block-wise post-training quantization (PTQ), which ignores the LLM head during quantization and minimizes the mean squared error (MSE) between the original and quantized model outputs, LFQ explicitly quantizes the LLM head by minimizing the cross-entropy between the token probability distributions of the original and quantized models. The authors apply LFQ to both general-purpose LLMs and reasoning-specialized models, such as DeepSeek-R1-Distill-Llama-8B, and evaluate the quantized models across multiple benchmarks, including language modeling, language understanding, instruction following, and mathematical reasoning tasks. Experimental results demonstrate that LFQ consistently improves the performance of quantized LLMs over existing block-wise PTQ methods.

**Strengths:**

1. LFQ can be easily integrated into various existing block-wise PTQ methods to further enhance the performance of quantized LLMs.

2. As shown in Table 2, the experimental results indicate that applying LFQ substantially enhances the performance of reasoning-specialized quantized LLMs on the challenging AIME 2024 benchmark when greedy decoding is used. This improvement is notable given the increasing prominence of reasoning-specialized LLMs.

**Weaknesses:**

1. The performance of LLMs appears to be primarily evaluated based on a single run using greedy decoding, which may compromise the reliability of the experimental results, as the responses of reasoning-specialized LLMs can deteriorate under greedy decoding [1]. To ensure a more robust evaluation, I recommend reporting the Avg@K metric, which averages performance scores over K independent runs under a common stochastic decoding setting (e.g., temperature = 0.6, top-p = 0.95).

2. The LLMs employed in the ablation study are not consistent with those used in the main experiments. Specifically, the ablation study is conducted exclusively on Llama-3.1-8B-Instruct, whereas the main experiments primarily involve Qwen2.5-7B-Instruct, Qwen2.5-14B-Instruct, L1-Qwen-7B-Max, and DeepSeek-R1-Distill-Llama-8B. This inconsistency undermines the comparability of the experimental results and may limit the validity of conclusions drawn from the ablation analysis.

[1] Incorrect Baseline Evaluations Call into Question Recent LLM-RL Claims

**Questions:**

Please see the Weaknesses.

---

> ### Author Response · Authors · 2025-11-21
> **Dear Reviewer nk9F,**
>
> Dear Reviewer nk9F,
>
> We sincerely appreciate your invaluable and constructive comments.
>
> --------------------------------
>
> $\textbf{[Weakness 1. Reporting the Avg@K metric.]}$
>
> Thank you for suggesting the Avg@K metric. First, we would like to gently remind the reviewer that, to verify that LFQ improves the generation quality of quantized reasoning-specialized LLMs not only under greedy decoding but also under stochastic decoding, we already reported Pass@8 on AIME’24 for L1-Qwen-7B-Max and DeepSeek-R1-Distill-Llama-8B with a temperature of 0.6 and a top-p of 0.95 in Table 2. Following the reviewer’s recommendation, we additionally measure Avg@8 on MATH500 for L1-Qwen-7B-Max and DeepSeek-R1-Distill-Llama-8B, again with a temperature of 0.6 and a top-p of 0.95, as shown in the table below.
>
>
> \<Table A. Avg@8 and standard deviation on MATH500 for L1-Qwen-7B-Max and DeepSeek-R1-Distill-Llama-8B with LFQ under block-wise PTQ (FlexRound). Within the PTQ method, the best accuracy is shown in $\textbf{bold}$. “W4” and “W3g128” denote 4-bit per-channel weight-only quantization and 3-bit group-wise quantization (group size $128$), respectively. We use a temperature of 0.6 and a top-p of 0.95.\>
>
> | Method | # Bits | MATH500 (Avg@8) |
> |:------------------------|:--------:|:--------------------------:|
> | L1-Qwen-7B-Max | BF16 | $89.05 \pm 0.74$ |
> | FlexRound | W4 | $87.45 \pm 0.75$ |
> | FlexRound + LFQ (Ours) | W4 | $\mathbf{88.40} \pm 0.86$ |
> | FlexRound | W3g128 | $85.35 \pm 0.64$ |
> | FlexRound + LFQ (Ours) | W3g128 | $\mathbf{86.50} \pm 0.54$ |
> ||||
> | DeepSeek-R1-Distill-Llama-8B | BF16 | $72.53 \pm 1.16$ |
> | FlexRound | W4 | $70.10 \pm 1.37$ |
> | FlexRound + LFQ (Ours) | W4 | $\mathbf{71.95} \pm 1.20$ |
> | FlexRound | W3g128 | $67.00 \pm 0.97$ |
> | FlexRound + LFQ (Ours) | W3g128 | $\mathbf{69.25} \pm 0.85$ |
> ||||
>
>
> Table A shows that LFQ also improves the Avg@K score on MATH500 across different reasoning-specialized LLMs, demonstrating that LFQ is effective under both greedy and stochastic decoding.
>
> We truly appreciate your constructive feedback for enhancing the clarity of our paper and have revised our paper to include this point and the above experimental results in Appendix C.
>
> --------------------------------
>
> $\textbf{[Weakness 2. Model inconsistency between main experiments and the ablation study.]}$
>
> Thank you for pointing this out. In line with the reviewer’s comment, we additionally conduct the ablation study for Qwen2.5-7B-Instruct as illustrated in the table below.
>
>
> \<Table B. Performance of Qwen2.5-7B-Instruct when block-wise PTQ methods (FlexRound, OmniQuant, and Block-AP) are incrementally augmented by (i) incorporating the LM head and (ii) using a logit-level cross-entropy objective in order to quantize the final Transformer block. Within each block-wise PTQ method, the best accuracy is shown in $\textbf{bold}$ and the second-best is $\underline{underlined}$. Here, all results use 4-bit per-channel weight-only quantization.\>
>
> | Method | LM-Head | Cross-Entropy | WikiText2 | MMLU | IFEval | MATH500 |
> |:-----------|:------------:|:--------------------:|:------------:|:---------:|:--------:|:--------------:|
> | Qwen2.5-7B-Instruct | N/A | N/A | $6.85$ | $73.49$ | $70.79$ | $74.2$ |
> ||||||||
> | FlexRound | X | X | $\underline{7.23}$ | $\mathbf{72.50}$ | $69.50$ | $\underline{72.6}$ |
> | FlexRound + LFQ | O | X | $7.26$ | $72.48$ | $\underline{71.35}$ | $71.4$ |
> | | O | O | $\mathbf{7.21}$ | $\underline{72.48}$ | $\mathbf{71.35}$ | $\mathbf{73.4}$ |
> ||||||||
> | OmniQuant | X | X | $7.73$ | $\underline{71.00}$ | $68.21$ | $69.8$ |
> | OmniQuant + LFQ | O | X | $\mathbf{7.29}$ | $\mathbf{71.02}$ | $\underline{68.95}$ | $\underline{70.6}$ |
> | | O | O | $\underline{7.53}$ | $70.99$ | $\mathbf{69.50}$ | $\mathbf{71.6}$ |
> ||||||||
> | Block-AP | X | X | $\underline{7.87}$ | $69.60$ | $66.73$ | $68.0$ |
> | Block-AP + LFQ | O | X | $7.92$ | $\underline{69.75}$ | $\underline{67.28}$ | $\underline{68.4}$ |
> | | O | O | $\mathbf{7.77}$ | $\mathbf{69.94}$ | $\mathbf{68.02}$ | $\mathbf{69.0}$ |
> ||||||||
>
>
> Table B further demonstrates that LFQ (ours, using both the LM head and cross-entropy) consistently improves generation quality across block-wise PTQ, while preserving the language modeling and understanding performance of existing methods. We therefore conclude that leveraging both the LM head and cross-entropy is essential for improving the generation quality of low-bit quantized LLMs.
>
> We greatly appreciate your valuable feedback for enhancing the clarity of our paper and have revised our paper to include this point and the above experimental results in Appendix A.
>
> --------------------------------
>
> Once again, we sincerely appreciate your time and efforts in reviewing our paper. If you have any remaining issues or concerns, please do not hesitate to bring them to our attention.

---

> > ### Author Response · Authors · 2025-11-27
> >
> > Dear Reviewer nk9F,
> >
> > We hope this message finds you well.
> >
> > We would like to sincerely thank you once again for the time and effort you have devoted to reviewing our paper. With about one week remaining before the deadline for reviewer–author discussions, we kindly wish to draw your attention to our response to your thoughtful comments. If you have any further questions or concerns, please do not hesitate to let us know.
> >
> > Best regards,
> >
> > Authors

---

> ### Author Response · Authors · 2025-12-02
>
> -------------------------
>
> $\textbf{[Weakness 1. Additional response on reporting the Avg@K metric]}$
>
> Beyond reasoning-specialized LLMs, to demonstrate that LFQ improves the generation quality of quantized instruction-tuned LLMs under both greedy and stochastic decoding, we additionally evaluated Avg@8 on MATH500 for Qwen2.5-7B-Instruct using a temperature of 0.6 and a top-p of 0.95.
>
> \<Table C. Avg@8 and standard deviation on MATH500 for Qwen2.5-7B-Instruct with LFQ under block-wise PTQ (FlexRound, OmniQuant, and Block-AP). Within each PTQ method, the best accuracy is shown in $\textbf{bold}$. “W4” and “W3g128” denote 4-bit per-channel weight-only quantization and 3-bit group-wise quantization (group size $128$), respectively. We use a temperature of 0.6 and a top-p of 0.95.\>
>
> | Method | # Bits | MATH500 (Avg@8) |
> |:------------------------|:--------:|:--------------------------:|
> | Qwen2.5-7B-Instruct | BF16 | $73.65 \pm 1.09$ |
> ||||
> | FlexRound | W4 | $69.33 \pm 1.21$ |
> | FlexRound + LFQ (Ours) | W4 | $\mathbf{70.35} \pm 1.16$ |
> | FlexRound | W3g128 | $63.83 \pm 0.92$ |
> | FlexRound + LFQ (Ours) | W3g128 | $\mathbf{65.35} \pm 0.76$ |
> ||||
> | OmniQuant | W4 | $68.28 \pm 1.00$ |
> | OmniQuant + LFQ (Ours) | W4 | $\mathbf{69.53} \pm 1.00$ |
> | OmniQuant | W3g128 | $61.75 \pm 1.12$ |
> | OmniQuant + LFQ (Ours) | W3g128 | $\mathbf{63.58} \pm 0.89$ |
> ||||
> | Block-AP | W4 | $65.98 \pm 1.42$ |
> | Block-AP + LFQ (Ours) | W4 | $\mathbf{67.25} \pm 1.15$ |
> | Block-AP | W3g128 | $58.08 \pm 0.82$ |
> | Block-AP + LFQ (Ours) | W3g128 | $\mathbf{61.53} \pm 1.25$ |
> ||||
>
>
> Table C further shows that LFQ improves Avg@K on MATH500 for both reasoning-specialized and instruction-tuned LLMs, demonstrating its effectiveness under both greedy and stochastic decoding.
>
> -------------------------
>
> $\textbf{[Additional experiments on MoE architectures]}$
>
> We conducted additional LFQ experiments on **Qwen3-30B-A3B-Instruct-2507**, one of the most powerful open-source MoE models, to highlight that LFQ is effective for both MoE models and traditional dense models.
>
> \<Table C. Performance of Qwen3-30B-A3B-Instruct-2507 with LFQ under block-wise PTQ (FlexRound). Within the block-wise PTQ setting, the best accuracy is shown in $\textbf{bold}$. “W4g128” denotes 4-bit group-wise quantization with a group size of 128. For Pass@1 evaluation on AIME$^{\prime}$25, we use a temperature of 0.6 and a top-p of 0.95, and sample 16 responses per question with a maximum generation length of 32768 tokens.\>
>
> | Method | # Bits | WikiText2 | AIME$^{\prime}$25 (Pass@1) |
> |:----------|:------:|:-----------:|:-----------:|
> | Qwen3-30B-A3B-Instruct-2507 | BF16 | $7.00$ | $62.50$ |
> | GPTQ | W4g128 | $7.32$ | $53.75$ |
> | Block-wise PTQ | W4g128 | $7.34$ | $55.21$ |
> | Block-wise PTQ + LFQ (Ours) | W4g128 | $\mathbf{7.29}$ | $\mathbf{59.38}$ |
> ||||
>
> The table above demonstrates that LFQ increases the Pass@1 score on AIME$^{\prime}$25 by more than 4 percentage points, thereby substantially reducing the performance gap between the quantized MoE model and its FP counterpart.
>
> -------------------------

---

### Official Review · Reviewer_ACQf · 2025-11-07

**Soundness:** 3
**Presentation:** 3
**Contribution:** 2
**Rating:** 6
**Confidence:** 1

**Summary:**

This paper introduces Logit-aware Final-block Quantization (LFQ), a method that enhances block-wise post-training quantization (PTQ) for large language models by aligning token probability distributions at the logit level. Unlike traditional block-wise PTQ methods that rely on mean squared error and neglect the LM head, LFQ minimizes cross-entropy between logits of the quantized and full-precision models to better preserve generative quality. Experiments on Llama 3.1 and 3.2 show LFQ consistently improves text generation performance while maintaining comparable language modeling and understanding accuracy to the full-precision baseline.

**Strengths:**

1. Clearly identifies and addresses a key limitation of existing PTQ methods (i.e., the mismatch between quantized and full-precision token distributions.)

2. Proposes a conceptually simple yet empirically robust method that yields consistent improvements across multiple architectures and benchmarks.

**Weaknesses:**

1. LFQ shows weaker performance compared to some baselines on certain benchmarks. For example, it underperforms LoRA-based quantization error compensation (RILQ) on some understanding tasks, indicating trade-offs in adaptation.

2. The work lacks analysis of computational overhead or convergence behavior during LFQ optimization, which could affect practical deployment scalability.

**Questions:**

could the method generalize to activation quantization or mixed-precision scenarios?

---

> ### Author Response · Authors · 2025-11-21
> **Dear Reviewer ACQf,**
>
> Dear Reviewer ACQf,
>
> We greatly appreciate your constructive and helpful feedback.
>
> --------------------------------
>
> $\textbf{[Weakness 1. LFQ underperforms LoRA-based quantization error compensation techniques (RILQ) on language modeling and understanding.]}$
>
> As the reviewer astutely observed, LFQ underperforms RILQ on WikiText2 perplexity (language modeling) and MMLU accuracy (language understanding), while outperforming RILQ on IFEval and GSM8K (text generation), as shown in Table 4. However, we emphasize that LFQ (quantization objective) and RILQ (LoRA addition) address orthogonal aspects of the problem rather than competing with each other. Because LFQ can be readily combined with RILQ in a complementary manner, we therefore explore their joint application to leverage the strengths of both methods.
>
>
>
> \<Table A. Comparison between RILQ, LFQ, and LFQ + RILQ for  Llama 3.1 8B Instruct using block-wise PTQ (FlexRound, OmniQuant, and Block-AP). Within each block-wise PTQ method, the best accuracy is shown in $\textbf{bold}$ and the second-best is $\underline{underlined}$. “W4” denotes 4-bit per-channel weight-only quantization.\>
>
> | Method | # Bits | WikiText2 | MMLU | IFEval | GSM8K |
> |:-----------|:-------:|:------------:|:---------:|:--------:|:--------------:|
> | Llama 3.1 8B Instruct | BF16 | $6.75$ | $68.34$ | $74.49$ | $84.99$ |
> |||||||
> | FlexRound | W4 | $7.06$ | $66.19$ | $70.24$ | $81.35$ |
> | FlexRound + RILQ | W4 | $\mathbf{6.95}$ | $66.86$ | $71.90$ | $80.52$ |
> | FlexRound + LFQ | W4 | $7.06$ | $\mathbf{66.97}$ | $\underline{72.09}$ | $\mathbf{81.80}$ |
> | FlexRound + LFQ + RILQ | W4 | $\underline{6.98}$ | $\underline{66.96}$ | $\mathbf{72.46}$ | $\underline{81.43}$ |
> |||||||
> | OmniQuant | W4 | $7.49$ | $64.87$ | $70.61$ | $78.17$ |
> | OmniQuant + RILQ | W4 | $\underline{7.24}$ | $\mathbf{66.07}$ | $71.35$ | $78.85$ |
> | OmniQuant + LFQ | W4 | $7.47$ | $65.48$ | $\underline{71.35}$ | $\mathbf{79.76}$ |
> | OmniQuant + LFQ + RILQ | W4 | $\mathbf{7.23}$ | $\underline{65.82}$ | $\mathbf{71.35}$ | $\underline{79.45}$ |
> |||||||
> | Block-AP | W4 | $7.76$ | $63.24$ | $68.58$ | $73.84$ |
> | Block-AP + RILQ | W4 | $\underline{7.43}$ | $\mathbf{64.62}$ | $68.58$ | $73.92$ |
> | Block-AP + LFQ | W4 | $7.69$ | $63.77$ | $\mathbf{68.76}$ | $\mathbf{74.45}$ |
> | Block-AP + LFQ + RILQ | W4 | $\mathbf{7.43}$ | $\underline{64.53}$ | $\underline{68.58}$ | $\underline{74.22}$ |
> |||||||
>
>
> As shown in Table A, LFQ + RILQ performs comparably to RILQ on WikiText2 perplexity and MMLU accuracy, while achieving results close to LFQ on IFEval and GSM8K. This indicates that LFQ + RILQ can effectively inherit the strengths of both techniques.
>
> We really appreciate your valuable feedback and have revised our manuscript to include this point and the above experimental results in Appendix E.
>
> --------------------------------
>
> $\textbf{[Weakness 2. Lack of computation overhead analysis.]}$
>
> Thank you for pointing this out. On a single A100 GPU, the LFQ process takes approximately 1.5 hours for Qwen2.5-7B-Instruct, L1-Qwen-7B-Max, DeepSeek-R1-Distill-Llama-8B, and Llama 3.1 8B Instruct, and about 2 hours for Qwen2.5-14B-Instruct.
>
> We sincerely appreciate your invaluable feedback for enhancing the clarity of our paper and have clarified this point in Appendix B.
>
> --------------------------------
>
> $\textbf{[Question 1. Generalization to activation quantization or mixed-precision configurations.]}$
>
> Thank you for the interesting suggestion. Since LFQ is designed for weight quantization as in Eq. (2) and agnostic to bit-width as seen in Table 1 and 2, it can be readily extended to mixed-precision weight quantization. However, additional modifications are required to generalize LFQ to activation quantization. Exploring such an extension is an interesting direction for future work.
>
> --------------------------------
>
> Once again, we sincerely appreciate your time and efforts in reviewing our paper. If you have any remaining issues or concerns, please do not hesitate to bring them to our attention.

---

> > ### Author Response · Authors · 2025-11-27
> >
> > Dear Reviewer ACQf,
> >
> > We hope this message finds you well.
> >
> > We would like to sincerely thank you once again for the time and effort you have devoted to reviewing our paper. With about one week remaining before the deadline for reviewer–author discussions, we kindly wish to draw your attention to our response to your thoughtful comments. If you have any further questions or concerns, please do not hesitate to let us know.
> >
> > Best regards,
> >
> > Authors

---

> ### Author Response · Authors · 2025-12-02
>
> -------------------------
>
> $\textbf{[Additional experiments on MoE architectures]}$
>
> We conducted additional LFQ experiments on **Qwen3-30B-A3B-Instruct-2507**, one of the most powerful open-source MoE models, to highlight that LFQ is effective for both MoE models and traditional dense models.
>
> \<Table C. Performance of Qwen3-30B-A3B-Instruct-2507 with LFQ under block-wise PTQ (FlexRound). Within the block-wise PTQ setting, the best accuracy is shown in $\textbf{bold}$. “W4g128” denotes 4-bit group-wise quantization with a group size of 128. For Pass@1 evaluation on AIME$^{\prime}$25, we use a temperature of 0.6 and a top-p of 0.95, and sample 16 responses per question with a maximum generation length of 32768 tokens.\>
>
> | Method | # Bits | WikiText2 | AIME$^{\prime}$25 (Pass@1) |
> |:----------|:------:|:-----------:|:-----------:|
> | Qwen3-30B-A3B-Instruct-2507 | BF16 | $7.00$ | $62.50$ |
> | GPTQ | W4g128 | $7.32$ | $53.75$ |
> | Block-wise PTQ | W4g128 | $7.34$ | $55.21$ |
> | Block-wise PTQ + LFQ (Ours) | W4g128 | $\mathbf{7.29}$ | $\mathbf{59.38}$ |
> ||||
>
> The table above demonstrates that LFQ increases the Pass@1 score on AIME$^{\prime}$25 by more than 4 percentage points, thereby substantially reducing the performance gap between the quantized MoE model and its FP counterpart.
>
> -------------------------

---

### Meta-Review · Area_Chair_CwmG · 2026-01-11

**Summary:**

This paper proposes a logit-aware final-block quantization objective to improve generation quality in low-bit PTQ models. Reviewers generally find the motivation clear and the method conceptually simple, with empirical gains on several generation benchmarks, while maintaining comparable performance on language modeling and understanding.

At the same time, multiple substantive concerns remain unresolved. Reviewers note trade-offs relative to LoRA-based error compensation on language modeling and understanding, limited analysis of optimization stability and convergence, and incomplete evidence regarding robustness under stochastic decoding across the full evaluation suite. Questions around generalization to other quantization settings (e.g., activation quantization or AWQ-style methods) and the sufficiency of empirical validation for key claims were not conclusively addressed, and no reviewer explicitly confirmed resolution of these issues post-rebuttal.

Overall, the reviews reflect mixed opinions with marginal scores and no explicit post-rebuttal score increases. Given the interrupted review process, this assessment represents a best-effort, conservative judgment based solely on the available reviews, rebuttal, and limited discussion, without assuming changes in reviewer positions.

**Reviewer Concerns:**

Reviewer ACQf:

Addressed:
• None.

Partially addressed:
• LFQ vs. RILQ trade-off: additional LFQ+RILQ results mitigate gaps, but LFQ-alone still underperforms on LM/understanding.
• Computational overhead: single-GPU runtime reported, but no deeper scalability or convergence analysis.

Outstanding:
• Convergence and stability analysis of LFQ optimization.
• Empirical validation for activation quantization / mixed-precision generalization.

Reviewer nk9F:

Addressed:
• Model inconsistency in ablations: added ablation on Qwen2.5-7B-Instruct aligned with main experiments.

Partially addressed:
• Robustness beyond greedy decoding: Avg@K added for selected tasks/models, but coverage remains limited.

Outstanding:
• Comprehensive stochastic-decoding robustness across the full benchmark suite.

Reviewer p8dA:

Addressed:
• Clarification of LM-head treatment: LM head is frozen in BF16 and used only for optimization signal.

Partially addressed:
• Applying LFQ to more layers: justification and limited k≤3 analysis provided, but no last-10-block experiments.

Outstanding:
• Adapting LFQ to AWQ: rationale given, but no direct LFQ-on-AWQ experiments.

**Reviewer Scores:**

Reviewer ACQf:

Original score: 6

Likely post-rebuttal score: 6

Justification:
• No explicit reviewer signal post-rebuttal.
• Outstanding major concerns on convergence/stability remain.

Reviewer nk9F:

Original score: 4

Likely post-rebuttal score: 4

Justification:
• No explicit reviewer signal post-rebuttal.
• Robustness under stochastic decoding not fully resolved.

Reviewer p8dA:

Original score: 4

Likely post-rebuttal score: 4

Justification:
• No explicit reviewer signal post-rebuttal.
• Main concern (LFQ adaptation to AWQ) remains outstanding.

---

### Decision · Program_Chairs · 2026-01-26

Reject